# 3-Bromopyruvate Suppresses the Malignant Phenotype of Vemurafenib-Resistant Melanoma Cells

**DOI:** 10.3390/ijms232415650

**Published:** 2022-12-09

**Authors:** Patrik da Silva Vital, Murilo Bonatelli, Marina Pereira Dias, Larissa Vedovato Vilela de Salis, Mariana Tomazini Pinto, Fátima Baltazar, Silvya Stuchi Maria-Engler, Céline Pinheiro

**Affiliations:** 1Molecular Oncology Research Center, Barretos Cancer Hospital, Barretos 14784-400, SP, Brazil; 2Barretos School of Health Sciences Dr. Paulo Prata—FACISB, Barretos 14785-002, SP, Brazil; 3Life and Health Sciences Research Institute (ICVS), School of Medicine, University of Minho, 4710-057 Braga, Portugal; 4ICVS/3B’s-PT Government Associate Laboratory, 4710-057 Braga, Portugal; 5Clinical Chemistry and Toxicology Department, School of Pharmaceutical Sciences, University of São Paulo, São Paulo 04023-901, SP, Brazil

**Keywords:** drug resistance, melanoma, metabolism, proto-oncogene protein B-raf, vemurafenib

## Abstract

(1) *BRAF* mutations are associated with high mortality and are a substantial factor in therapeutic decisions. Therapies targeting *BRAF*-mutated tumors, such as vemurafenib (PLX), have significantly improved the overall survival of melanoma patients. However, patient relapse and low response rates remain challenging, even with contemporary therapeutic alternatives. Highly proliferative tumors often rely on glycolysis to sustain their aggressive phenotype. 3-bromopyruvate (3BP) is a promising glycolysis inhibitor reported to mitigate resistance in tumors. This study aimed to evaluate the potential of 3BP as an antineoplastic agent for PLX-resistant melanoma treatment. (2) The effect of 3BP alone or in combination with PLX on viability, proliferation, colony formation, cell death, migration, invasion, epithelial-mesenchymal marker and metabolic protein expression, extracellular glucose and lactate, and reactive species were evaluated in two PLX-resistant melanoma cell lines. (3) 3BP treatment, which was more effective as monotherapy than combined with PLX, disturbed the metabolic and epithelial-mesenchymal profile of PLX-resistant cells, impairing their proliferation, migration, and invasion and triggering cell death. (4) 3BP monotherapy is a potent metabolic-disrupting agent against PLX-resistant melanomas, supporting the suppression of the malignant phenotype in this type of neoplasia.

## 1. Introduction

Melanoma is the deadliest form of skin cancer [1]. The incidence of this neoplasia rises yearly, reaching 324,635 cases, with 57,043 deaths globally in 2020 [2]. Although tumor resection is generally problem solving for early stages, the advanced appearances of disease often require complex approaches [3], such as detailed screening for frequent oncogenic variants [4]. The mutated *BRAF* gene is dominant among melanoma patients, occurring in approximately 50% of cases [5] and rendering a worse prognosis [6]. Drugs targeting BRAF mutations approved by the Food and Drug Administration agency (FDA), such as vemurafenib, have culminated in drastic improvement in the response rates and overall survival of melanoma patients. However, patient relapse within a year of treatment remains a problem [7]. Currently, combined BRAF/MEK [8,9,10] and immunological checkpoint inhibition [11,12,13], which are encountered even in adjuvant [14] and neoadjuvant [15] contexts, are flagships for unresectable melanomas. Nevertheless, targeted therapy resistance patterns similar to those with monotherapy [16], low response rates, and slow onset found with immunotherapies [11,12,13] remain obstacles in melanoma treatment.

Similar to other types of cancer [17], metabolic plasticity in melanomas is an essential factor in therapeutic response and resistance mechanisms [18]. *BRAF*-mutated melanomas often undergo metabolic changes toward aerobic glycolysis to supply their demands for growth and division [19]. Since these alterations make them more susceptible to BRAF inhibitors [20], resistant melanomas reprogram their metabolism toward oxidative phosphorylation in response to glycolysis impairment induced by BRAF inhibition [18], increasing their mitochondrial biomass and activity [21,22], as well as enhancing anaplerotic reactions, e.g., the glutamine pathway [23,24]. Altogether, these data suggest that glycolysis might be an essential factor in tumoral survival, whereas mitochondrial metabolism might correlate with tumoral resistance adaptation and development [18]. Interestingly, a central intermediary in both glycolytic and oxidative pathways, either as a final product or substrate, is lactate [25]—a metabolite that, in turn, requires monocarboxylate transporters (such as MCT1 and MCT4) to cross membranes [26]. The overexpression of these transporters is associated with worse prognosis in patients [27] and with several processes during tumor progression, including increased proliferation, angiogenesis, metastasis, and drug resistance [28,29,30]. As a result, MTC1 inhibition is offering promising results, as demonstrated by a clinical trial supporting its role as a potential therapeutic target for various types of cancer [31]. MCT1 has also been described as a possible biomarker for tumor sensitivity to 3-bromopyruvate (3BP) [32], a halogenated pyruvate analog that can trigger a metabolic catastrophe by inhibiting glycolysis and glutaminolysis-related proteins, targeting mitochondrial metabolism, and causing oxidative stress [33]. However, despite this broad action, specific characteristics make 3BP a promising drug for highly proliferative tumors. These characteristics include its stability and facilitated uptake by MCT1 in the acidic extracellular microenvironment of tumors [32], its ability to inhibit tumor-specific overexpressed enzymes [34], its potential to imbalance free radicals [35,36], and its ability to resensitize chemoresistant tumors [37,38]. However, despite promising results found in case reports [39], as well as in vitro and in vivo studies [33,40], no clinical trials are in progress using 3BP.

Therefore, to explore possible therapeutic alternatives for melanoma treatment, this study aimed to evaluate the effect of 3BP as an antineoplastic agent in PLX-resistant melanomas.

## 2. Results

### 2.1. A375R and SKMEL28R Cells Present Different Metabolic Profiles with Different Sensitivities to 3-Bromopyruvate 

To investigate possible differences in the metabolic profiles of melanoma cell lines, Western blot analysis of proteins related to glycolysis and glutaminolysis pathways was performed (Figure 1a,b). Both cell lines presented high expression of MCT1 and HK2. Comparative analysis also indicated that A375R cells had increased expression of GLUD1/2, GLUT1, and PDK, while GLS, LDH5, and MCT4 showed higher expression in the SKMEL28R cell line. In summary, these data suggest that the SKMEL28R cell line exhibits a diversified metabolic profile with no clear predominance of glycolysis or the glutaminolysis pathway, whereas the A375R cell line apparently presents more glycolytic behavior.

The effect of 3BP on the viability of PLX-resistant melanoma cell lines after 24 h of treatment was assessed in the presence and absence of PLX (Figure 1c). Since these results were similar after 48 h of treatment (Appendix A), the IC_50_ values (60 µM for A375R and 80 µM for SKMEL28R in 3BP monotherapy and 45 µM for A375R and 90 µM for SKMEL28R in 3BP/PLX combined therapy) and IC_25_ values (45 µM for A375R and 65 µM for SKMEL28R in 3BP monotherapy and 35 µM for A375R and 80 µM for SKMEL28R in 3BP/PLX combined therapy) obtained after 24 h of treatment were used for the functional assays. A PLX viability curve comparing resistant and naïve cells confirmed the resistant phenotype of the studied melanoma cells (Appendix A). Overall, the comparative analysis between the IC_50_ values of both studied PLX-resistant melanoma cell lines indicated that A375R cells were more responsive to 3BP treatment, regardless of PLX presence (Figure 1d).

### 2.2. 3-Bromopyruvate Modulates the Metabolic Profile of Vemurafenib-Resistant Melanoma Cells

To check possible alterations in the metabolic profile caused by 3BP treatment, Western blot analysis of metabolic proteins was performed after 24 h of treatment (Figure 2). The results showed that, in the absence of PLX, 3BP significantly decreased LDH5 expression in both PLX-resistant melanoma cell lines, whereas it increased MCT1 expression in A375R cells and HK2 expression in SKMEL28R cells. Additionally, 3BP increased GLS, GLUT1, and MCT4 expression in a PLX-dependent manner and upregulated CA9, regardless of PLX presence, in the SKMEL28R cell line. However, to verify whether 3BP alters glucose consumption or lactate production, the extracellular amounts of those metabolites were quantified.

Notably, despite decreasing glucose uptake in SKMEL28R cells, 3BP treatment significantly stimulated the influx of that metabolite in A375R cells after 4 h of treatment (Figure 3a and Appendix A). The last response was consistent after 24 h, with 3BP exposure significantly increasing glucose consumption in both cell lines (Figure 3b). Conversely, lactate production remained constant in A375R cells but was significantly decreased in SKMEL28R cells after 3BP treatment in a PLX-independent manner (Figure 3c).

### 2.3. 3-Bromopyruvate Decreases Cell Division and Induces Cell Death in Vemurafenib-Resistant Melanoma Cells

To determine whether the viability impairment caused by 3BP was due to the decline in cell division, a proliferation assay was performed (Figure 4a). The results showed that 3BP decreases cell proliferation in a PLX-independent manner in both the A375R and SKMEL28R cell lines. Supporting these findings, 3BP treatment also significantly restrained the ability of the cell lines to form colonies regardless of the presence of PLX (Figure 4b,c). To determine whether cell death also contributed to the viability decrease caused by 3BP, annexin V and propidium iodide assays were performed (Figure 4d and Appendix A). The results showed that 3BP stimulated A375R and SKMEL28R cell death, independently of PLX. Notably, 3BP monotherapy exhibited a more pronounced effect on A375R cell death than combined therapy. Additionally, 3BP also promoted late apoptosis/necrosis in the SKMEL28R cell line. In an attempt to address other mechanisms of action of 3BP, reactive oxygen species (ROS) release was also evaluated. A significant increase in A375R cells producing ROS was observed after 3BP treatment, although similar results were not observed in the SKMEL28R cell line (Figure 4e and Appendix A).

### 2.4. 3-Bromopyruvate Impairs the Migration and Invasion of Vemurafenib-Resistant Melanoma Cells and Inhibits the Expression of Epithelial-Mesenchymal Transition Markers

Further investigations of the effect of 3BP were performed concerning the migration and invasion of PLX-resistant melanoma cells (Figure 5). The results showed that 3BP was able to impair invasion in both studied cell lines regardless of PLX presence. Conversely, although migration was restrained in SKMEL28R cells regardless of PLX presence, 3BP did not affect A375R cell migratory potential in the presence of PLX. To avoid the likely interference of the high percentage of cell death generated by 3BP in the A375R cell line, the IC_25_ was also used in migration and invasion assays, providing similar results to IC_50_. To evaluate the reasons for these results, Western blot analysis was performed to assess the expression of epithelial-mesenchymal transition (EMT) markers. As depicted in Figure 6, 3BP alone decreased the expression of ASMA, ECAD, and SLUG in A375R cells and increased the expression of SNAIL in SKMEL28R cells. In contrast, 3BP combined with PLX increased VIM expression in A375R cells and decreased SLUG expression in SKMEL28R cells. In addition, 3BP impaired NCAD and TGFB expression regardless of the presence of PLX in A375R and SKMEL28R cells, respectively.

## 3. Discussion

Tumors present high bioenergetic and biosynthetic demands [41]. For this reason, they are commonly highly dependent on glycolysis [42] and glutaminolysis [43]. These demands are even more prominent in chemoresistant cells since they require additional resources to manage chemotherapeutic stress [37,38]. Both PLX, a drug able to decrease proliferation [44], and 3BP, a potent Warburg effect inhibitor [33], act upon different pathways of the known hallmarks of cancer [45]. Therefore, taking advantage of the interactions between oncogenic and metabolic pathways [46], we evaluated whether the combination regime PLX/3BP might be an interesting anticancer strategy. In general, 3BP monotherapy has shown promising results in arresting melanoma aggressiveness, especially in highly glycolytic cell lines. To our knowledge, this study is the first addressing the combination of these drugs in PLX-resistant melanoma cell lines.

Melanoma is one of the most heterogeneous types of cancer [47], presenting a high metabolic diversity, which can impact clinical response and generate drug resistance [48]. Therefore, to extrapolate the obtained results to tumors with different metabolic profiles, metabolic characterization was performed to ensure the metabolic diversity of the cells used in the present study. Comparatively, the A375R cell line exhibited considerably low expression of GLS (suggesting a low glutaminolytic activity) and upregulated glycolysis-related proteins, such as GLUT1 and PDK. In contrast, the SKMEL28R cell line showed no predominance on glycolytic or glutaminolytic pathways, albeit presenting a high comparative expression of LDH5 and MCT4. These metabolic differences seem to have contributed to greater sensitivity of A375R to 3BP compared to SKMEL28R; relative IC_50_ values among those cell lines were 1.37-fold for 3BP monotherapy and 1.85-fold for 3BP/PLX combined therapy. Importantly, although A375R cells exhibited a lower IC_50_ than SKMEL28R cells, both cell lines were responsive to 3BP treatment in the functional assays. This response corroborates the protein expression characterization data, which showed a high expression of MCT1 in those cell lines. This transporter and its chaperone CD147 are dominant 3BP uptake mechanisms that can modulate cell sensitivity to this drug [32]. High MCT1 expression has also been associated with an increase in the metastatic potential of melanomas, strongly suggesting that this transporter might be an interesting biomarker for tumor aggressiveness [49] and therefore justifying the use of 3BP in this neoplasia.

Interestingly, 3BP decreased proliferation and colony formation, in addition to triggering cell death regardless of PLX presence, indicating that 3BP presents cytostatic and cytotoxic effects on melanoma cells. These results agree with the literature, which shows that 3BP can both arrest the cell cycle [50,51,52] and trigger apoptosis/necrosis [53,54] in various types of tumors. However, as concentrations of 3BP greater than 60 μM can completely deplete cellular ATP [55], we hypothesized that necrosis might be the principal mechanism of 3BP in the SKMEL28R cell line. In fact, an increased percentage of cells in late apoptosis/necrosis was found in this cell line after 3BP treatment, supporting these data. Conversely, A375, which did not present the same pattern of cell death, significantly increased the amount of ROS after 3BP treatment. Since one mechanism of action of 3BP is the production of free radicals by inhibition of SDH/complex II [56] and depletion of free GSH [35,36], this result might indicate that, instead of necrosis, 3BP is generating cell death in this cell line primarily by oxidative stress. Interestingly, the effect of 3BP on cell viability was fully reversed after NAC exposure (Appendix A), showing that, although 3BP might induce cell death by different mechanisms, free radical formation is an essential mechanism for both cell lines.

Metabolic reprogramming controls EMT [57], an event that contributes to the aggressive phenotype of cancer cells and is directly influenced by the MAPK pathway [58]. As a glycolytic inhibitor, 3BP is a potential EMT suppressor [59], in addition to being reported to inhibit relevant proteins for the invasive phenotype, such as matrix metalloproteinases (MMPs) [60]. In accordance with these data, in the present study, 3BP treatment not only drastically decreased the migratory and invasive potential of PLX-resistant cell lines but also induced an expression change in their EMT profiles, as indicated by alteration in the expression of EMT markers [58,61]. Several pro-mesenchymal factors were downregulated after 3BP treatment, including the mesenchymal markers ASMA and NCAD, the transcription factor SLUG, and the EMT-promoter TGFB. However, this process might not be compensation free since 3BP also downregulated the epithelial marker ECAD in the absence of PLX and positively expressed the mesenchymal marker VIM and transcription factor SNAIL in the presence of PLX. Since EMT inhibition is not a binary event, these compensatory signals indicate that EMT inhibition by 3BP might be a partial event [61]. In fact, given that SNAIL and SLUG might be associated with different stages of invasion, SLUG inhibition might reflect the disruption of migratory maintenance caused by 3BP, whereas SNAIL upregulation might be a compensatory effect to this response [62]. Notably, since ECAD can trigger the MAPK pathway through EGFR activation, its inhibition might also explain the effect of 3BP in reducing aggressiveness in the absence of PLX [63,64]. However, it is worth mentioning that downregulation of ECAD and upregulation of SNAIL and VIM are associated with EMT stress adaptation in esophageal and breast cancer [65,66]. Therefore, although EMT inhibition might be an important mechanism of action by which 3BP impairs the maintenance of the melanoma invasive/migratory phenotype, this process might be partial or gradual [61] and suggests that possible clinical uses of this drug should be applied with caution.

Studies have indicated that prolonged exposure to BRAF/MEK inhibitors renders melanomas less dependent on glycolysis but addicted to mitochondrial metabolism and anaplerotic pathways, such as glutaminolysis [21,22]. The increased expression of glycolytic proteins, such as GLUT1, MCT4, CA9, and LDH5, after removing PLX from the culture media might corroborate this point. Since 3BP is a primordial glycolysis inhibitor [33], the metabolic shift might be responsible for the greater effectivity of 3BP monotherapy over combined therapy. In fact, given that PLX can also inhibit glycolysis [18], this responsibility might be the cause of the competitive interaction between PLX and 3BP observed in the combination assays (Appendix A). Additionally, glutaminolysis stimulation by PLX exposure can interfere with ROS production (one of the mechanisms of action of 3BP), further impairing 3BP activity [67]. In contrast, the A375R cell line, which notably expressed more glycolytic enzymes after PLX removal, had its proliferation, cell death, and DHE assays more affected by 3BP monotherapy. In fact, in contrast with 3BP/PLX combined treatment, which produced no significant outcomes, 3BP alone stopped A375R cell migration even upon IC_25_ treatment (indicating low interference of cell death in this process). Since the same was not encountered in SKMEL28R, it is hypothesized that complex metabolic/proliferative interactions should be occurring.

Remarkably, 4 h of treatment with 3BP increased glucose uptake in A375R cells. Although the same outcome was not found in SKMEL28R cells, 24 h of exposure to 3BP decreased extracellular glucose in both cell lines. These responses contrast with the inhibitory activity already described for this drug on HK2 [33], an enzyme that facilitates the metabolization of glucose inside the cell [68]. Nonetheless, studies have indicated that, although high HK2 expression can potentiate the 3BP effect, HK2 silencing is not sufficient to alter the sensitivity to this drug [69]. Additionally, the nonconsensus regarding the effects of 3BP on HK2 [70] and the existence of compensatory isoforms [71] might indicate that other factors are involved in 3BP activity on this enzyme. In fact, SKMEL28R presented increased HK2 expression after 3BP treatment, a response that corroborates the increased glucose consumption after exposure to this compound. Moreover, studies undertaken with metformin (an inhibitor of mitochondrial complex II) showed similar results, attributing the proliferation impairment and increased glucose consumption to the coordinator of energetic homeostasis with AMPK [72,73]. Treatment with 3BP also decreased extracellular lactate in the SKMEL28R cell line regardless of PLX presence. This result, already described in the literature [33], agrees with the downregulation of LDH5 observed in this cell line, suggesting that 3BP treatment was able to decrease lactate production in the studied cell line. However, evidence by others has pointed out that, although lactate export might be affected by 3BP, a steady increase in this metabolite was observed in the intracellular environment after exposure to this compound, likely occasioned by the impairment of lactate transport across membranes [70]. MCT1 and MCT4 levels were not impaired in the present study. Instead, 3BP treatment increased the expression of these transporters, together with several metabolic enzymes, including CA9, GLS, GLUT1, and HK2. In agreement with the increase in glucose uptake found in these cell lines, a hypothesis is that, as SKMEL28R is exposed to 3BP, it increases efficiency in cell metabolism to overcome the acute stress rendered by this drug. A significant factor in this context is the AMPK-HIF1A axis, which is stabilized by ROS balance [74] and is involved in the regulation of several proteins that are overexpressed by 3BP [75]. In fact, even CA9, an important hypoxic marker [76,77], was upregulated by 3BP regardless of the PLX presence in this cell line, indicating that more studies are required to unravel the nuances behind this mechanism.

As a limitation in this study, it is of paramount importance to emphasize that neither studied cell line was obtained from PLX-resistant melanoma samples. Since PLX resistance was achieved by continuous exposure to PLX from established cell lines, this inhibitor was maintained in the culture medium throughout the entire cell culture period to avoid acquired resistance issues, such as resistance loss at the time of the experiments. That continuous exposure seems to have contributed to a more aggressive phenotype in the control group (Dulbecco’s phosphate-buffered saline; DPBS) after 24 h of PLX absence indicates that those cells did not become addicted to this drug. This effect, observed in the increase in viability in the absence of PLX (likely due to intensified proliferation), contrasts with the literature that claims that drug holidays in melanoma tumor cells with acquired resistance to PLX cause a growth disadvantage upon drug withdrawal, leading to tumor regression [78]. These results suggest either that primary cells might be a more feasible model to study PLX addiction or that a prolonged incubation period without PLX exposure might be necessary for the cell line to manifest its dependency on this drug. Notably, although drug resensitization after drug holidays has been found in vitro, in vivo studies still struggle to demonstrate this effect [79], demonstrating that more investigations are necessary to determine the relevant aspects regarding this topic.

## 4. Materials and Methods

### 4.1. Cell Lines and Growth Conditions

A375 melanoma naïve cells were kindly donated by Keiran Smalley (Moffitt Cancer Center, Tampa, FL, USA), and SKMEL28 melanoma naïve cells were donated by Dr. Marisol Soengas (Centro Nacional de Investigaciones Oncológicas, Madrid, Spain). Melanoma cells were maintained in Dulbecco’s modified Eagle’s medium (DMEM; Sigma, St. Louis, MI, USA) with 4500 mg/L glucose and 4 mM L-glutamine supplemented with 10% fetal bovine serum (FBS; Gibco, Middlesex County, MA, USA) and 1% penicillin-streptomycin solution (Sigma Aldrich, St. Louis, MI, USA) at 37 °C and 5% CO_2_. Their PLX-resistant counterparts, A375R and SKMEL28R, were established as previously described [80,81,82]. In short, naïve A375 and SKMEL28 cell lines were plated onto 60 mm plates at a density of 1 × 10^4^ cells and treated with 0.5–6 µM PLX every 3 days for 4–6 weeks. Clonal colonies were then isolated, and PLX (4.5 µM for A375R and 6.0 µM for SKMEL28R) was replenished every 2–3 days to sustain their resistance phenotype. Short tandem repeat (STR) analysis was performed to ensure cell authenticity according to international standards [83]. Biweekly tests confirmed the cell lines’ free mycoplasma status (MycoAlert PLUS Mycoplasma Detection Kit; Lonza, Basel, Switzerland).

### 4.2. Drugs

3-bromopyruvate (Sigma Aldrich) was dissolved in DPBS (Cytiva, Marlborough, MA, USA) at room temperature immediately before use. Vemurafenib (Selleck Chem, Houston, TX, USA) was dissolved in dimethyl sulfoxide (DMSO; Sigma Aldrich) to prepare stock solutions of 10 µM, which were stored at 4 °C until use.

### 4.3. Cell Viability and IC_50_ and IC_25_ Determination

Cells were plated onto 96-well plates at a density of 5 × 10^3^ cells/well. Then, the cells were exposed for 24 h or 48 h to increasing concentrations of PLX (0, 2, 4, 6, 8, 10, 12, 14 µM) or 3BP (0, 5, 10, 25, 50, 75, 100, and 150 µM) in the presence or absence of PLX. Subsequently, cell viability was measured by sulforhodamine B (SRB; Sigma Aldrich) assay, according to the manufacturer’s instructions. In short, cells were fixed with 50% trichloroacetic acid (Sigma Aldrich) for 1 h at 4 °C. Then, the cells were stained with 0.4% SRB for 30 min and washed with 1% acetic acid (Merck, Darmstadt, Germany). Finally, the incorporated stain was diluted in 10 mM Tris (Sigma Aldrich) and read by a spectrophotometer (Varioskan Flash; Thermo Fisher Scientific, Waltham, MA, USA) at 565 nm (λref = 690 nm). For IC_50_ and IC_25_ determination, the results from the cell viability assay were normalized by the viability percentage of the drug vehicle. GraphPad Prism software, version 8.0.1 (GraphPad Software, San Diego, CA, USA), was used for nonlinear regression analysis and the dose–response sigmoidal equation (variable slope). The obtained IC_50_ and IC_25_ values were used in the functional assays.

### 4.4. Drug Combined Effect

The combined effect of PLX and 3BP was evaluated by CalcuSyn software, version 2.0 (BioSoft, Cambridge, UK) based on the Chou–Talalay method. The effect of each drug on cell viability was calculated by SRB assay as follows: 3BP alone (range between 5 and 150 µM), PLX alone (range between 2 and 14 µM), 3BP + PLX combination for A375 and A375R (fixed 4.5 µM concentration for PLX and range between 5 and 150 µM for 3BP), and 3BP + PLX combination for SKMEL28 and SKMEL28R (fixed 6.0 µM concentration for PLX and range between 5 and 150 µM for 3BP). The resulting combination index (CI), calculated by the software, was used to determine drug interactions, namely additive effect (CI = 1), synergism (CI < 1), and antagonism (CI > 1) [84].

### 4.5. Cell Proliferation Assay

Cell proliferation was determined by 5-bromo-2-deoxyuridine (BrdU; Roche, Mannheim, Germany) incorporation according to the manufacturer’s instructions. In brief, the cells were plated onto a 96-well plate at a density of 5 × 10^3^ cells/well and allowed to grow until 80% confluence. Then, the cells were maintained under treatment conditions with 10% BrdU for 24 h. Subsequently, the cells were incubated in a DNA denaturant for 30 min, and the peroxidase-conjugated anti-BrdU antibody was then added and incubated for 90 min. Next, the cells were held in substrate solution for 30 min, and the final solution was analyzed by a spectrophotometer (Varioskan Flash; Thermo Fisher Scientific) at 370 nm (λref = 492 nm). The relative proliferation of the cells was determined as a percentage of the proliferation of the DPBS group.

### 4.6. Colony Formation Assay

Cells were plated onto 6-well plates at a density of 5 × 10^3^ cells/well and exposed to different treatment conditions in DMEM supplemented with 10% FBS for 20 days. The colonies formed were stained with crystal violet in 5% paraformaldehyde and photographed (SZX7 Stereomicroscope System; Olympus Corporation, Shinjuku City, Tokyo, Japan). Colony formation capacity was evaluated by solubilizing stained colonies in acetic acid and analyzing the resulting solution in a spectrophotometer (Varioskan Flash; Thermo Fisher Scientific) at 570 nm. The relative colony formation capacity was determined as a percentage of the absorbance of the DPBS group.

### 4.7. Cell Death Assay

Cell death was evaluated by annexin V and propidium iodide assays (Roche) according to the manufacturer’s instructions. In short, cells were plated onto 6-well plates at a density of 3 × 10^5^ cells/well, allowed to grow until 80% confluence, and exposed to treatment conditions for 24 h. Subsequently, both the supernatant and adherent cells were collected, washed in DPBS and incubated with annexin V and propidium iodide for 15 min at room temperature in the dark. The percentage of early or late apoptotic cells was determined by flow cytometry (Accuri C6 Plus Flow Cytometer; BD Biosciences, Franklin Lakes, NJ, USA), collecting 50,000 events for each condition.

### 4.8. Reactive Oxygen Species Quantification

Cells were plated onto 6-well plates at a density of 6 × 10^5^ cells/well, allowed to grow until 80% confluence, and then exposed to different treatment conditions for 24 h. Then, both adherent and supernatant cells were collected and treated with 5 µM dihydroethidium (DHE; Thermo Fisher Scientific) for 1 h and 30 min at room temperature while protected from light. Concomitantly, the negative control group was treated with 20 nm N-acetylcysteine (NAC; Sigma Aldrich). One hour after exposure to DHE, the positive control group was treated with antimycin A (AA, Sigma Aldrich). Subsequently, the cells were centrifuged and homogenized in DBPS. The percentage of positive DHE cells was evaluated by flow cytometry (Accuri C6 Plus Flow Cytometer; BD Biosciences), collecting 50,000 events for each condition.

### 4.9. Cell Migration and Invasion Assays

For the migration assay, 5 × 10^4^ cells were plated onto 24-well 0.4-µm Cell Culture Inserts (Corning, NY, USA). For the invasion assay, 5 × 10^5^ cells were plated onto previously rehydrated 24-well 0.8-µm Biocoat Matrigel Invasion Chambers (Corning, NY, USA). The cells were maintained under treatment conditions on the upper chamber in culture medium without FBS and allowed to migrate/invade for 24 h to the lower portion of the insert, which contained culture medium with 10% FBS. Migrating/invading cells were then fixed in methanol and stained with hematoxylin/eosin. Membranes were photographed (BX43 Light Microscope; Olympus Corporation), and migratory/invasive cells were counted by OpenCFU software, version 3.8 (Quentin Geissmann, Berlin, Germany). Relative cell migration/invasion was determined as a percentage of the number of migratory cells in the DPBS group.

### 4.10. Western Blot

Cells were plated at a density of 6 × 10^5^ cells/well onto 6-well plates, allowed to grow until 80% confluence, and then exposed to different treatment conditions for 24 h. Then, both adherent and supernatant cells were homogenized in cold lysis buffer (supplemented with protease inhibitors; Complete Protease Inhibitor Cocktail, Roche) for 15 min and centrifuged at 13,000 rpm for 15 min at 4 °C. The supernatant was collected, and the soluble protein concentration was quantified by Bradford (BioRad, Hercules, CA, USA). Aliquots containing 20 µg of total protein were separated on polyacrylamide gels by SDS–PAGE and transferred onto nitrocellulose membranes (Amersham Protran; GE Healthcare Life Sciences, Chicago, IL, USA) in 25 mM Tris-base/glycine buffer using the mini TransBlot Turbo Transfer System (BioRad). The membranes were blocked with 5% milk powder in TBS/0.1% Tween (TBST; pH 7.6) for 1 h at room temperature and then incubated in primary antibody overnight at 4 °C. The antibodies used were GLS (HPA036223, 1:125) from Atlas Antibodies (Stockholm, Sweden); CA9 (ab15086, 1:2000), GLUT1 (ab15309, 1:1000), HK2 (ab104836, 1:1000) and LDH5 (ab101562, 1:3000) from Abcam (Cambridge, United Kingdom); ASMA (#14968, 1:1000), ECAD (#3195S, 1:1000), GLUD1/2 (#12793, 1:1000), NCAD (#4061S, 1:250), SLUG (#9585, 1:1000), SNAIL (#3879, 1:1000), TGFB (#3711, 1:1000), and VIM (#5741S, 1:1000) from Cell Signaling Technology (Danvers, MA, USA); ASCT2 (abn73, 1:1000) from Merck Millipore (Burlington, MA, USA); and CD147 (sc-71038, 1:250), MCT1 (sc-365501, 1:200), MCT4 (sc-50329, 1:2000) and PDK (sc-28278, 1:2000) from Santa Cruz Biotechnology (Dallas, TX, USA). Subsequently, the membranes were washed in TBS-T and incubated in either anti-mouse (sc-2031, Santa Cruz Biotechnology) or anti-rabbit (sc-2020, Santa Cruz Biotechnology) antibody diluted 1:5000 in 5% milk powder in TBST. The blots were detected using chemiluminescence with either SignalFire ECL Reagent (Cell Signaling Technology) or SuperSignal West Femto Maximum Sensitivity Substrate (Thermo Fisher Scientific). The chemiluminescent signal was detected using ImageQuant LAS 4000 mini (GE Healthcare Life Sciences), and densitometry analysis was performed using ImageJ software, version 1.4, (National Institutes of Health, Bethesda, MD, USA). Β-actin (#3700, 1:1000) from Cell Signaling Technology was used as the loading control.

### 4.11. Extracellular Glucose and Lactate Quantification

Cells were plated onto 96-well plates at a density of 5 × 10^3^ cells/well and allowed to grow until 80% confluence. Then, cell media before (T_0_) and after 24 h (T_24_) of exposure to different treatment conditions were collected and stored at –20 °C. Glucose Colorimetric Assay and Lactate Colorimetric Assay (Spinreact, Girona, Spain) were used to quantify glucose and lactate, respectively, according to the manufacturer’s instructions. The formula T_0_-T_24_ was used to calculate the extracellular glucose variation, while the extracellular lactate variation was obtained from the formula T_24_-T_0_. The obtained values were normalized to the total biomass, which was determined by a cell viability assay. Data are expressed as total µmol of metabolite/total biomass.

### 4.12. Glucose Uptake Quantification

Cells were plated onto 6-well plates at a density of 6 × 10^5^ cells/well and allowed to grow until 80% confluence. Then, the cells were exposed to 10 µM 2-(N-(7-nitrobenz-2-oxa-1,3-diazol-4-yl)amino)-2-deoxyglucose (2NBDG, Thermo Fisher Scientific) in DMEM without glucose under treatment conditions for 4 h. Cells were treated with 5 µM apigenin as a positive control. After treatment, both adherent and suspended cells were collected, washed in DPBS, and exposed to 5 µM 7-aminoactinomycin D (7AAD; BD Biosciences) for 15 min at room temperature protected from light. The determination of viable and positive 2NBDG cell percentages was evaluated by flow cytometry (Accuri C6 Plus Flow Cytometer—BD Biosciences), collecting 50,000 events for each condition.

### 4.13. Statistical Analysis

GraphPad Prism software, version 8.0.1, was utilized for statistical analysis. Student’s *t*-test was used to evaluate significant differences between DPBS and 3BP and between the DPBS + PLX and 3BP + PLX groups, as well as significant differences between cell lines, considering *p*-values ≤ 0.05 significant.

## 5. Conclusions

The present study provides insights into the possible mechanisms of action by which an antimetabolic agent acts on PLX-resistant melanoma cell lines. It was found that 3BP can decrease proliferation, induce necrosis, generate ROS, and dysregulate EMT proteins in PLX-resistant melanomas. However, a metabolic characterization was shown to be a primordial step in screening patients who might benefit from 3BP therapy. Factors such as MCT1 expression and glycolytic activity seem to contribute to 3BP sensitivity, but evidence indicates that glutaminolytic activity and antioxidant capacity might be disadvantageous for 3BP activity. According to the obtained results, 3BP and PLX present a competitive interaction. Nevertheless, PLX-resistant melanomas were shown to be sensitive to 3BP monotherapy, posing it as a promising alternative for this type of cancer.

## Figures and Tables

**Figure 1 ijms-23-15650-f001:**
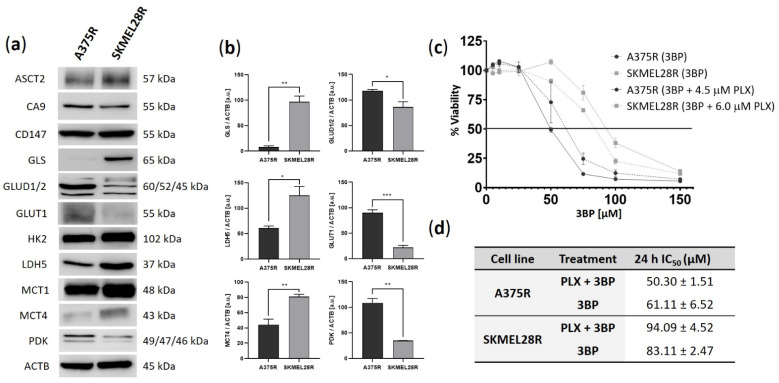
Basal expression of metabolic proteins and cell viability analysis using a sulforhodamine B (SRB) assay in vemurafenib (PLX)-resistant melanoma cells treated with 3-bromopyruvate (3BP) for 24 h. (**a**) Western blot analysis of the basal levels of ASCT2, CA9, CD147, GLS, GLUD1/2, GLUT1, HK2, LDH5, MCT1, MCT4, and PDK in PLX-resistant melanoma cells. (**b**) Quantification of immunoblots normalized to ACTB levels. The *y*-axis represents arbitrary units [a.u.]. Data represent the mean ± SEM of at least three independent experiments. (**c**) The dose–response curve represents the viability of PLX-resistant melanoma cells treated with 0–150 µM 3BP, either alone or combined with PLX (4.5 µM for A375R cells and 6.0 µM for SKMEL28R cells). Data were normalized to the vehicle control (100%). (**d**) IC_50_ values of PLX-resistant melanoma cells treated with 3BP for 24 h. Data represent the mean ± SEM of at least three independent experiments performed in triplicate. * *p* ≤ 0.05; ** *p* ≤ 0.01; *** *p* ≤ 0.001.

**Figure 2 ijms-23-15650-f002:**
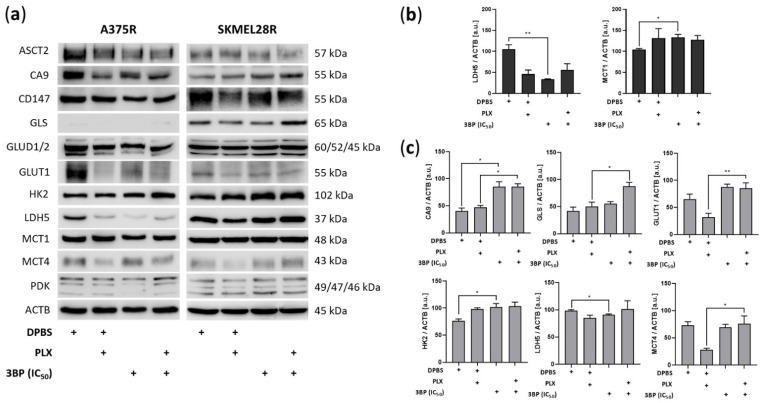
Expression of metabolic proteins in vemurafenib (PLX)-resistant melanoma cells treated with 3-bromopyruvate (3BP) for 24 h. (**a**) Western blot analysis of the levels of ASCT2, CA9, CD147, GLS, GLUD1/2, GLUT1, HK2, LDH5, MCT1, MCT4, and PDK in PLX-resistant melanoma cells treated with IC_50_ 3BP, either alone or combined with PLX (4.5 µM for A375R and 6.0 µM for SKMEL28R), for 24 h. (**b**) Quantification of immunoblots normalized to ACTB levels in A375R melanoma cells. (**c**) Quantification of immunoblots normalized to ACTB levels in SKMEL28R melanoma cells. The *y*-axis represents arbitrary units [a.u.]. The positive signs in the bar charts indicate the presence of a defined treatment. Data represent the mean ± SEM of at least three independent experiments. * *p* ≤ 0.05; ** *p* ≤ 0.01.

**Figure 3 ijms-23-15650-f003:**
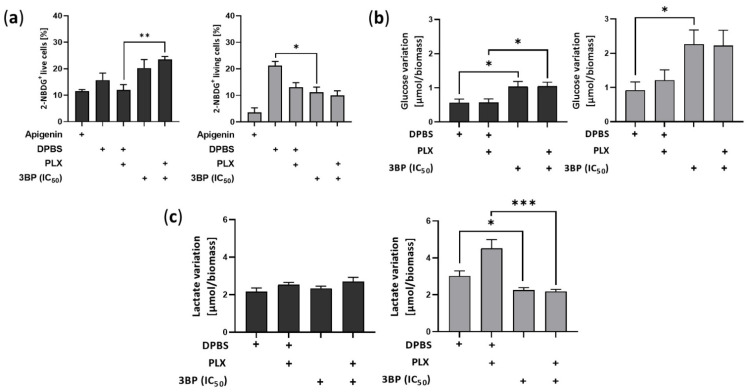
Analysis of variation in extracellular glucose and lactate levels in vemurafenib (PLX)-resistant melanoma cells treated with 3-bromopyruvate (3BP). PLX-resistant melanoma cells were treated with IC_50_ 3BP, either alone or combined with PLX (4.5 µM for A375R cells and 6.0 µM for SKMEL28R cells). (**a**) Glucose uptake in PLX-resistant melanoma cells treated for 4 h after 7-aminoactinomycin D (7AAD)/2-(N-(7-nitrobenz-2-oxa-1,3-diazol-4-yl)amino)-2-deoxyglucose (2NBDG) staining. Data represent the percentage of total cell counts of 7AAD− and 2NBDG+; (**b**) glucose variation (influx) in PLX-resistant melanoma cells treated for 24 h. Data were calculated by the difference between µmol of glucose in culture media at 0 h and after 24 h of treatment. Data were normalized by cell viability analysis using a sulforhodamine B (SRB) assay. (**c**) Lactate variation (efflux) in PLX-resistant melanoma cells treated for 24 h. Data were calculated by the difference between µmol of lactate in culture media after 24 h of treatment and at 0 h. Data were normalized by cell viability analysis using a sulforhodamine B (SRB) assay. Dark gray represents A375R cells; light gray represents SKMEL28R cells. The positive signs in the bar charts indicate the presence of a defined treatment. Data represent the mean ± SEM of at least three independent experiments performed in triplicate. * *p* ≤ 0.05; ** *p* ≤ 0.01; *** *p* ≤ 0.001.

**Figure 4 ijms-23-15650-f004:**
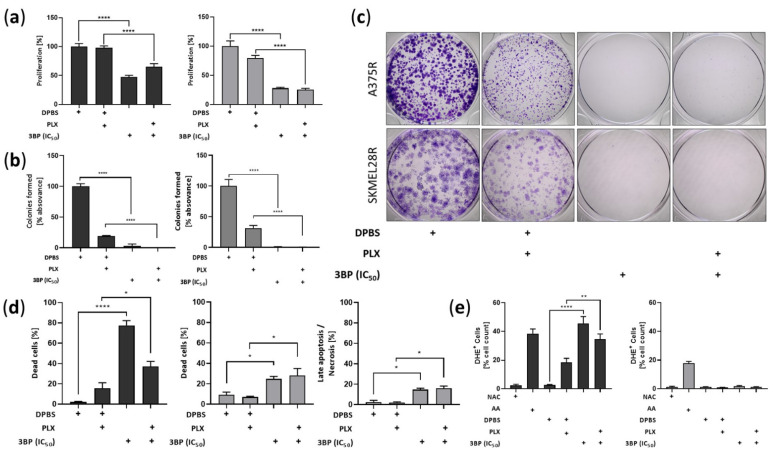
Proliferation, colony formation, cell death, and reactive oxygen species (ROS) production analysis in vemurafenib (PLX)-resistant melanoma cells treated with 3-bromopyruvate (3BP) for 24 h. PLX-resistant melanoma cells were treated with IC_50_ 3BP, either alone or combined with PLX (4.5 µM for A375R and 6.0 µM for SKMEL28R), for 24 h. (**a**) Cell proliferation analysis using the 5-bromo-2-deoxyuridine (BrdU) assay. Data were normalized to the vehicle control (100%); (**b**) colony formation analysis in PLX-resistant melanoma cells after violet crystal staining. Graphic representation of absorbance after solubilization of stained cell colonies in acetic acid. Data were normalized to the vehicle control (100%); (**c**) representative PLX-resistant melanoma cell colonies stained with crystal violet; (**d**) cell death analysis using annexin V/propidium iodide (PI) staining. Data representing the statistically significant results obtained by the cell death assay. Early apoptosis: percentage of total cell count annexin V+ and PI−. Late apoptosis/necrosis: percentage of total cell count annexin V− and PI+; (**e**) free radical formation analysis using dihydroethidium (DHE) staining. AA (antimycin A) was used as the positive control; NAC (N-acetylcysteine) was used as the negative control. Data are presented as the percentages of the total DHE+ cell count. Dark gray represents A375R cells; light gray represents SKMEL28R cells. The positive signs in the bar charts indicate the presence of a defined treatment. Data represent the mean ± SEM of at least three independent experiments performed in triplicate. * *p* ≤ 0.05; ** *p* ≤ 0.01; **** *p* ≤ 0.0001.

**Figure 5 ijms-23-15650-f005:**
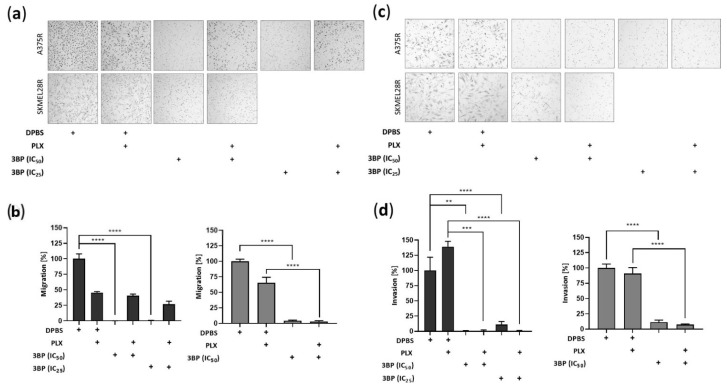
Migration and invasion analysis using the transwell assay in vemurafenib (PLX)-resistant melanoma cells treated with 3-bromopyruvate (3BP) for 24 h. PLX-resistant melanoma cells were treated with IC_25_ or IC_50_ 3BP, either alone or in combination with PLX (4.5 µM for A375R cells and 6.0 µM for SKMEL28R cells), for 24 h. (**a**) representative migratory PLX-resistant melanoma cells stained with hematoxylin and eosin; (**b**) graphic representation of migratory cell count; (**c**) representative invasive PLX-resistant melanoma cells stained with hematoxylin and eosin; (**d**) graphic representation of invasive cell count. Data were normalized to the vehicle control (100%). Dark gray represents A375R cells; light gray represents SKMEL28R cells. The positive signs in the bar charts indicate the presence of a defined treatment. Data represent the mean ± SEM of at least three independent experiments performed in triplicate. ** *p* ≤ 0.01; *** *p* ≤ 0.001; **** *p* ≤ 0.0001.

**Figure 6 ijms-23-15650-f006:**
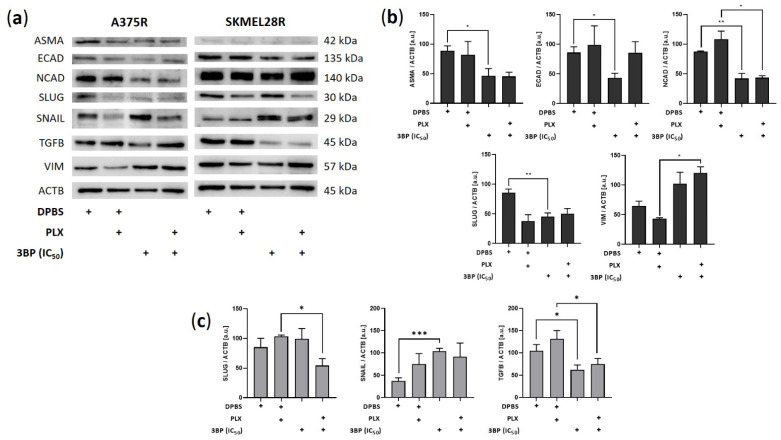
Expression of epithelial-mesenchymal transition (EMT) markers in vemurafenib (PLX)-resistant melanoma cells treated with 3BP for 24 h. (**a**) Western blot analysis of the levels of ASMA, ECAD, NCAD, SLUG, SNAIL, TGFB, and VIM in PLX-resistant melanoma cells treated with IC_50_ 3-bromopyruvate (3BP), either alone or combined with PLX (4.5 µM for A375R cells and 6.0 µM for SKMEL28R cells), for 24 h. (**b**) Quantification of immunoblots normalized to ACTB levels in A375R melanoma cells. (**c**) Quantification of immunoblots normalized to ACTB levels in SKMEL28R melanoma cells. The *y*-axis represents arbitrary units [a.u.]. The positive signs in the bar charts indicate the presence of a defined treatment. Data represent the mean ± SEM of at least three independent experiments. * *p* ≤ 0.05; ** *p* ≤ 0.01; *** *p* ≤ 0.001.

## Data Availability

The data generated in this study are available within the article and its Appendix A.

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
