# Peer review of "3-Bromopyruvate Suppresses the Malignant Phenotype of Vemurafenib-Resistant Melanoma Cells"

_ijms, 2022, doi:10.3390/ijms232415650_

Round 1

Reviewer 1 Report

While targeted therapeutics brought hope for melanoma patients, the development of resistance is still a serious clinical challenge. da Silva Vital et al., analyzed activity of 3-bromopuryvate against two vemurafenib-resistant melanoma cell lines. Impressive number of proteins have been checked in WB experiments. While results describing a drug with potential activities against resistant melanomas would certainly add interest in the field, there are several issues the authors should address.   

1.      More details should be provided about the method of obtaining PLX-resistant cell lines (how they were established).  

2.      What does it mean: “… not presenting a specific cell death pattern on flow cytometry analysis”? Original data (histograms) showing results of Annexin V/PI staining should be included in the Supplementary materials.

3.      What does it mean “triplicates” in case of WB results?

4.      I would recommend a more rigorous description and discussion of the data. For example antagonistic effects that have been detected for PLX/3BP combination should be better discussed.

5.      It might be helpful to provide a scheme summarizing results.

6.      Heavy editorial work would be necessary as the text needs substantial rewriting and restructuring, as well as checking and querying of the consistency. Several sentences are not clear e.g., “Overall, the IC50 values showed that the A375R cell line was more sensitive to 3BP, alone or combined therapy with PLX”: (1) the sensitivity of A375R cells to 3BP was compared to …? (2) grammar should  be improved.

Other examples of unclear expressions:

a)      Data normalized by the viability percentage of the drug vehicle.

b)     Melanoma cell lines IC50 values …

c)      IC50 comparation etc.

In the present form the manuscript is very difficult to follow and should be definitely improved.

Author Response

Response to Reviewer 1 Comments

Point 1: More details should be provided about the method of obtaining PLX-resistant cell lines (how they were established).

Response 1: As recommended, we provided a more detailed description of how cell resistance was established (item 4.1).

Point 2: What does it mean: “… not presenting a specific cell death pattern on flow cytometry analysis”? Original data (histograms) showing results of Annexin V/PI staining should be included in the Supplementary materials.

Response 2: As suggested, we included original data and graphics summarizing cell death pattern distribution in the Supplementary materials (Figure S4). We also added these data for 2NDBG (Figure S3) and DHE (Figure S5) assays. With "Not presenting a specific cell death", we meant that the A375R cell, unlike SKMEL28R, did not show significant alteration in early or late apoptosis/necrosis. To clarify, we changed the mentioned sentence to "In fact, an increased percentage of cells in late apoptosis/necrosis was found in this cell line after 3BP treatment, supporting this data. In fact, an increased percentage of cells in late apoptosis/necrosis was found in this cell line after 3BP treatment, supporting these data. Conversely, A375, which did not present the same pattern of cell death, significantly increased the amount of ROS after 3BP treatment" (lines 274-278).

Point 3: What does it mean “triplicates” in case of WB results?

Response 3: The term "triplicates" relating to WB was incorrect. The WB analyses were performed in three independent experiments only. Thus, all expressions "triplicates" were removed from this context.

Point 4: I would recommend a more rigorous description and discussion of the data. For example antagonistic effects that have been detected for PLX/3BP combination should be better discussed.

Response 4: Besides PLX inhibiting the same pathway that 3BP acts upon 3BP (glycolysis), we added the hypothesis that PLX and 3BP have antagonistic interactions because PLX long exposure enhances glutaminolysis, a pathway that influences 3BP activity by quenching ROS production. Modifications in the manuscript include:

"Studies have indicated that prolonged exposure to BRAF/MEK inhibitors renders melanomas less dependent on glycolysis but addicted to mitochondrial metabolism and anaplerotic pathways, such as glutaminolysis." (lines 310-312)

" In fact, given that PLX can also inhibit glycolysis, this responsibility might be the cause of the competitive interaction between PLX and 3BP observed in the combination assays. " (lines 316-318)

"Additionally, glutaminolysis stimulation by PLX exposure can interfere with ROS pro-duction (one of the mechanisms of action of 3BP), further impairing 3BP activity." (lines 318-320).

We also added a scheme in the graphic summary to explain that hypothesis.

Point 5: It might be helpful to provide a scheme summarizing results.

Response 5: We agree that a scheme summarizing the main findings extensively will help the general understanding of the work. Thus, we drew a graphic summary to be included at the end of the paper.

Point 6: Heavy editorial work would be necessary as the text needs substantial rewriting and restructuring, as well as checking and querying of the consistency. Several sentences are not clear e.g., “Overall, the IC50 values showed that the A375R cell line was more sensitive to 3BP, alone or combined therapy with PLX”: (1) the sensitivity of A375R cells to 3BP was compared to …? (2) grammar should be improved.

Other examples of unclear expressions:

  1. a) Data normalized by the viability percentage of the drug vehicle.
  2. b) Melanoma cell lines IC50 values …
  3. c) IC50 comparation etc.

In the present form the manuscript is very difficult to follow and should be definitely improved.

Response 6: Thank you for the advice! All mentioned sentences were restructured. Additionally, after detecting grammar issues in figure descriptions, we restructured all figure legends of the work. We also submit the manuscript to professional English revision (Please see the attachment).

Reviewer 2 Report

The manuscript titled “3-bromopyruvate suppresses the malignant phenotype of 2 vemurafenib-resistant melanoma cells” by Vital et al. describes the role of 3BP in vemurafenib-resistant melanoma cells. The manuscript needs to be mechanistically improved and there is lack of in vivo experiment to demonstrate the efficacy of 3-BP. In addition, there are several comments that need to be addressed:

1.     Fig 1: The authors must compare MCT1 and HK2 expression with the respective parental cell lines? Proper controls are missing in these experiments.

2.     I don’t think the IC50 values for A375R are significant in 3BP alone vs the combined treatment group. The authors needs to perform an invivo experiment to check the efficacy of 3 BP?

3.     The manuscript needs to written more scientifically for example some areas such as figure legends needs to be modified.

4.     The effect of 3BP on cellular apoptosis needs to be explored further. For example, what is the effect of 3BP on apoptotic markers? It needs to be supported using further experiment. In addition, just showing ROS induction/quenching in A375R alone does not exactly explore the role of 3BP in ROS induced DNA damage signing in this cell lines. Further mechanisms are necessary to confirm the exact role of 3BP.

5.     I feel It is necessary to compare the effect of 3BP in parental versus resistant cell lines to confirm the therapeutic role of 3BP instead.

6.     Figure 5A: The migration images are not great quality. It has to be substituted with better quality images. The western blot quality can be improved.

7.     In addition, further experiments are necessary to establish the exact role of 3BP. The authors have tried to show the role of 3BP in cell death, ROS induced signaling or EMT, But has not fully explored any single pathway completely.

Author Response

Response to Reviewer 2 Comments

Point 1: The authors must compare MCT1 and HK2 expression with the respective parental cell lines? Proper controls are missing in these experiments.

Response 1: The basal WB analysis in this manuscript was performed to compare the different metabolic profiles of the PLX-resistant melanoma cells under study. These data were necessary to explain the differences in the 3BP response and sensibility encountered in those cells. As our aim was to focus on the effect of 3BP on vemurafenib resistance, basal WB analysis on naive cell lines was not performed.

Point 2: I don’t think the IC50 values for A375R are significant in 3BP alone vs the combined treatment group. The authors needs to perform an invivo experiment to check the efficacy of 3 BP?

Response 2: In fact, the results presented in the manuscript indicate that the difference between 3BP alone vs 3BP/PLX combined is not significant for viability assay. However, that difference becomes clearer for other parameters such as cell death, ROS production, and, especially for migration assay. Therefore, aspiring on how antimetabolic agents could be applied in clinical practice, we decided to highlight monotherapy over combined therapy. In vivo experiments are currently being performed in our research group to check 3BP efficacy and hopefully will be published soon in another manuscript.

Point 3: The manuscript needs to written more scientifically for example some areas such as figure legends needs to be modified.

Response 3: Thank you for the advice! We rewrote all the Figure Legends in the manuscript to ensure they will be more clear and more understandable to read. We also submit the manuscript to professional English revision (Please see the attachment).

Point 4: The effect of 3BP on cellular apoptosis needs to be explored further. For example, what is the effect of 3BP on apoptotic markers? It needs to be supported using further experiment. In addition, just showing ROS induction/quenching in A375R alone does not exactly explore the role of 3BP in ROS induced DNA damage signing in this cell lines. Further mechanisms are necessary to confirm the exact role of 3BP.

Response 4: Most of the effects of 3BP upon cell death and ROS formation are already described by other authors. During the process of performing experiments, we detected that 3BP treatment significantly reduced migration and invasion of PLX-resistant melanoma cells. Hence, we decided to further investigate a hypothesis that 3BP, as an antimetabolic agent, could impair EMT. The idea surrounding metabolic pathways influence EMT is already established. However, to our knowledge, that is the first work exploring the role of 3BP in this process. Those are the reasons that motivated us to check EMT markers by WB analysis, but not other markers related to cell death.

Point 5: I feel It is necessary to compare the effect of 3BP in parental versus resistant cell lines to confirm the therapeutic role of 3BP instead.

Response 5: We analyzed the effect of PLX, 3BP, and PLX+3BP on the cell viability of naive and PLX-resistant A375 and SKMEL28 melanoma cells. Those data can be found in Figure S2.

Point 6: Figure 5A: The migration images are not great quality. It has to be substituted with better quality images. The western blot quality can be improved.

Response 6: As recommended by the reviewer, we improved the quality of the mentioned figures by enhancing the size and converting them to TIFF format.

Point 7: In addition, further experiments are necessary to establish the exact role of 3BP. The authors have tried to show the role of 3BP in cell death, ROS induced signaling or EMT, But has not fully explored any single pathway completely.

Response 7: We completely agree that further experiments are necessary to fully understand the role of 3BP on PLX-resistant melanomas. As 3BP can act upon several different pathways, this compound can provide an almost illimited source of studies for several fields. Our focus in this manuscript, however, was on the metabolic alterations caused by this compound in this type of neoplasia. Hence, we explored the expression of several proteins related to the most usual metabolic pathways in cancer resistance (glycolysis and glutaminolysis), providing useful information on how this type of compound acts in this context.

Round 2

Reviewer 2 Report

The authors have clarified all the comments raised in the manuscript and the manuscript can be accepted.